# Encapsulation of AgNPs in a Lignin Isocyanate Film: Characterization and Antimicrobial Properties

**DOI:** 10.3390/ma16124271

**Published:** 2023-06-08

**Authors:** Edwin S. Madivoli, Sammy I. Wanakai, Pius K. Kairigo, Rechab S. Odhiambo

**Affiliations:** 1Chemistry Department, Jomo Kenyatta University of Agriculture and Technology, Nairobi P.O. Box 62000-00200, Kenya; 2Department of Biological and Environmental Science, University of Jyvaskyla, P.O. Box 35, FI-40014 Jyvaskyla, Finland; 3Department of Physical Science, University of Kabianga, Kericho P.O. Box 2030-20200, Kenya

**Keywords:** lignin, silver nanoparticles, polyurethane films

## Abstract

Lignin isolated from agricultural residues is a promising alternative for petroleum-based polymers as feedstocks in development of antimicrobial materials. A polymer blend based on silver nanoparticles and lignin–toluene diisocyanate film (AgNPs–Lg–TDIs) was generated from organosolv lignin and silver nanoparticles (AgNPs). Lignin was isolated from *Parthenium hysterophorus* using acidified methanol and used to synthesize lignin capped silver nanoparticles. Lignin–toluene diisocyanate film (Lg–TDI) was prepared by treating lignin (Lg) with toluene diisocyanate (TDI) followed by solvent casting to form films. Functional groups present and thermal properties of the films were evaluated using Fourier-transform infrared spectrophotometry (FT–IR), thermal gravimetry (TGA), and differential scanning calorimetry (DSC). Scanning electron microscopy (SEM), UV–visible spectrophotometry (UV–Vis), and Powder X-ray diffractometry (XRD) were used to assess the morphology, optical properties, and crystallinity of the films. Embedding AgNPs in the Lg–TDI films increased the thermal stability and the residual ash during thermal analysis, and the presence of powder diffraction peaks at 2θ = 20, 38, 44, 55, and 58⁰ in the films correspond to lignin and silver crystal planes (111). SEM micrographs of the films revealed the presence of AgNPs in the TDI matrix with variable sizes of between 50 to 250 nm. The doped films had a UV radiation cut-off at 400 nm as compared to that of undoped films, but they did not exhibit significant antimicrobial activity against selected microorganisms.

## 1. Introduction

Agricultural residues are important materials that are receiving considerable attention since they may serve as feedstocks that can power several industries including, but not limited to, the manufacturing of composite materials [1,2,3]. Lignin, a byproduct of the paper and pulp industry, is normally used to produce heat during the pulping process (Figure 1), yet as an aromatic biopolymer, it can be used as the starting material for producing value-added materials such as vanillin [4,5]. It is isolated from agricultural residues using Kraft process [6], soda process [7], and/or extraction with ionic liquids. These processes are either expensive, or require high pressures, temperatures, and extreme pH which can lead to modification of the primary lignin structure [5,8,9,10]. Lignin is isolated from different plant samples which contain complex polymers that need harsh or mild solvents for depolymerization such acid hydrolysis, alkaline hydrolysis, enzyme, reductive or oxidative fractionation, and combined pretreatment methods (Figure 2) [11,12].

On the other hand, the organosolv process allows the delignification of lignocellulosic biomass in an environmentally benign manner with little alterations to lignins structure [13,14,15,16]. Because of its renewability, abundance, low cost, and presence of a plethora of functional groups, it is considered to be a promising biobased polyol substitute for polyurethane (PU) production [17]. This is made possible by the abundance of hydroxyl functional groups that are found in its phenylpropanoic structure, which make it a good candidate for production of polymer films and blends [4,18]. In this regard, lignin has been used to replace conventional polyols in the development of polyurethane films (PUFs) [2]. Fractionated lignin from switch grass, without modification, was utilized in the development of semirigid polyurethane foams [19]. A biobased PUF was also prepared by blending castor oil and Kraft lignin, in which the reported PUF had variable mechanical properties dependent on the lignin content. By incorporating lignin into the PUF, the glass transition temperature, the crosslinking density, and the mechanical stress of the PUF were ultimately improved [18,20,21]. Polyurethane films prepared using lignin have also been shown to have a high UV absorbance, good mechanical properties, and high thermal stability. Moreover, incorporation of metal nanoparticles into the composite materials has been shown to increase the antimicrobial capacity of the resultant composite material [17,22,23]. In this regard, green synthesized metallic nanoparticles such as AgNPs offer great potential, especially if they have been synthesized in an environmental benign way. Due to their anti-inflammatory, antifungal, and antiplatelet activities, silver nanoparticles are well suited for incorporating into composite materials as they display a broad spectrum of antimicrobial activity [24,25,26]. Therefore, the effect of incorporating lignin and AgNPs on the morphology, thermal, mechanical, and antimicrobial properties of AgNPs–Lg–TDI composite films were investigated. The functional groups present, morphology, and thermal properties of the films were investigated using a Fourier-transform infrared spectrophotometer, scanning electron microscope, differential scanning calorimetry, and thermal gravimetric analyzer. Powder X-ray diffraction was used to evaluate the crystallinity of the films before and after embedding silver nanoparticles. A disc diffusion assay was used to evaluate the antimicrobial properties of the polymer films against *Staphylococcus aureus* (ATCC-25923), *Bacillus subtilis*, *Pseudomonas aeruginosa* (ATCC-27853), *Escherichia coli* (ATCC-25922), and *Candida albicans* (ATCC 90028).

## 2. Materials and Methods

### 2.1. Materials

Analytical-grade silver nitrate (ACS reagent), methanol (99.5% purity), sulfuric acid, and TDI (mw = 174.16) were purchased from Sigma Aldrich and used as received.

### 2.2. Extraction of Lignin

Lignin was isolated from *Parthenium hysterophorus* using acidified methanol as an extracting solvent as previously described (Figure 3) [20,21,27].

In brief, one hundred grams of ground *Parthenium hysterophorus* were weighed into a one-liter flask followed by the addition of 1500 mL of acidified methanol which was prepared by addition of 1% of H_2_SO_4_ as a catalyst into pure methanol [20,21]. The reaction media was heated at 50 °C for 10 h after which the lignin present in the extracting solvent was concentrated in a Buchi rotary evaporator (Büchi Labortechnik AG, Flawi, Switzerland) where the solvent used was recovered. In order to separate the hydrolyzed lignin from nonhydrolyzed lignin, a 1:1 *v*/*v* of hexane ethyl acetate mixture was used to separate the mixture in a separating funnel. After separation, the nonhydrolyzed lignin (20 g) present in the organic phase was concentrated and used for the synthesis of lignin–toluene diisocyanate composite films [21].

### 2.3. Synthesis of Silver Nanoparticles–Lignin–Toluene Diisocyanate Films (AgNPs–Lg–TDI)

Silver nanoparticles used in this study were synthesized using lignin following a previously reported protocol [26,27]. In brief, 3 mL of AgNO_3_ (0.01 M) solution was mixed with 3 mL lignin (10% *m*/*v*) in a round bottom flask which was then placed in the dark for 2 h to allow the bioreduction of Ag^+^ to AgNPs to occur. The reaction was monitored using a Shimadzu UV–Vis spectrophotometer (Shimadzu, Kyoto, Japan) in the range of 300–800 nm and the reaction was deemed complete when the plasmon resonance peak of AgNPs was observed at 423 nm and the color of the solution changed to brick red. The silver nanoparticles–lignin–toluene diisocyanate (Lg–TDI) and silver–lignin–TDI films (AgNPs–Lg–TDI) were prepared by mixing lignin and TDI in the ratio 6:1 m/m to give composite films with 20% lignin at room temperature, followed by addition of AgNPs, and after polymerization, the solution was poured into a glass Petri dish and cured for 24 h in an oven (Bioevopeak Shandong, Jinan, China) set at 50 °C [4,28,29].

## 3. Characterization of AgNPs–Lg–TDI

### 3.1. Optical Properties

Optical properties of TDI, Lg–TDI, and AgNPs–Lg–TDI were determined using a Shimadzu UV–Vis 1800 spectrophotometer in the range of 200–800 nm [30]. The cast films were removed from the mold after curing, mounted in the spectrophotometer, and optical properties were evaluated by acquiring the spectra in transmittance mode.

### 3.2. Functional Group Analysis

A Bruker Tensor II Fourier-transform infrared spectrophotometer (Bruker, Ettlingen, Germany) was used to evaluate the functional groups present in the films. The thin films were mounted in the IR sample holders, and the spectra were acquired after setting spectral resolution at 4 cm^–1^ and the scanning range between 500 to 4000 cm^–1^ [31].

### 3.3. Crystal Structure Determination

The thin films were mounted on XRD sample holders and their XRD profile obtained using a STOE STADIP P X-ray Powder Diffraction System (STOE and Cie GmbH, Darmstadt, Germany). The X-ray generator was equipped with a copper tube operating at 40 kV and 40 mA and irradiating the sample with a monochromatic CuKα radiation with a wavelength of 0.1545 nm. The XRD spectra was acquired at room temperature over the 2θ range of 2–90° at 0.05° intervals with a measurement time of 1 s per 2θ intervals [32].

### 3.4. Thermal Profile

To study the effect of incorporating lignin on the thermal profile of the film, differential scanning calorimetry (DSC) and thermal gravimetric analysis were performed on a Mettler Toledo DSC/TGA 3^+^ system (Mettler-Toledo GmbH, Switzerland). When compared to cellulose and hemicellulose, lignin decomposes slower and over a broader temperature range between 200–500 °C; hence, 5 mg of the samples were heated in 40 µL aluminum crucibles from 25–500 °C at 10 °C min^−1^ and cooled to 25 °C at −22 °C min^−1^ [30,31,33].

### 3.5. Surface Analysis

The morphology of the films was evaluated using a scanning electron microscope (FEI XL30 Sirion FEG (Oxford Instruments Plc, Abingdon, UK)) equipped with an energy-dispersive x-ray spectrometer (EDS) system from EDAX having a lithium-doped silicon detector [34]. The thin films were mounted on carbon-coated SEM grid, gold-sputtered before analysis to prevent charging, and analyzed with the SEM operated at an accelerating voltage of 30 kV. The size distribution of the nanoparticles was obtained using image J software version 1.53u where sample points of the nanoparticles in the images were selected and their means, frequency, and histograms calculated [35].

### 3.6. Antimicrobial Assay

The antimicrobial properties of the samples were conducted using disc diffusion assay in which 6 mm discs of AgNPs–Lg–TDI and TDI were prepared using a disc punch and their antimicrobial activity evaluated against standard antibiotic samples. The microbial strains *Staphylococcus aureus* (ATCC-25923), *Bacillus subtilis*, *Pseudomonas aeruginosa* (ATCC-27853), *Escherichia coli* (ATCC-25922), and *Candida albicans* (ATCC 90028) were obtained from the Department of Medical Microbiology of the Jomo Kenyatta University of Agriculture and Technology, Kenya. The isolates were first subcultured in a nutrient broth (Oxoid) and incubated at 37 °C for 18 h while the fungal isolates were subcultured on a Sabouraud dextrose agar (SDA) (Oxoid) for 72 h at 25 °C. Bacterial cultures were adjusted to 0.5 McFarland turbidity standards and inoculated onto MHA (Oxoid) plates (diameter: 15 cm). Antibiotic discs of Amoxyclav (20/10 µg), Nitrofurantoin (200 µg), Nalidixic acid (30 µg), Gentamicin (10 µg), Norfloxacin (10 µg), Ofloxacin (10 µg), Ceftriaxone (30 µg), and Sulphamethoxazole (25 µg) were used as positive control while sterilized paper discs were used as negative controls for both the bacteria and fungi. The plates were then incubated for 24 h at 37 °C for bacteria and at 28 °C for fungi, and the antimicrobial activity was measured as inhibition zone around each disc. The experiments were performed in triplicate and the results expressed as mean ± standard deviations [36].

## 4. Results and Discussion

### 4.1. Synthesis of AgNPs–Lg

Figure 4 and Figure 5 depict UV–Vis spectra of lignin, lignin-capped silver nanoparticles, TDI, Lg–TDI, and AgNPs–Lg–TDI films.

Synthesis of AgNPs was followed by observation of their SPR peak between 500–600 nm using a UV–Vis spectrophotometer. From Figure 4, comparison between the spectra of lignin and AgNPs–Lg revealed that Lg was utilized as a reductant during synthesis of AgNPs, which was evident from the formation of the AgNPs SPR peak at 443 nm. It has been reported by several authors that this peak, which normally appears at variable wavelengths between 400–600 nm, is indicative of the presence of silver nanoparticles in the reaction media and the variability in its maximum wavelength is as a result of variable sizes [17,37]. Figure 5 shows the UV–Vis spectra of TDI, Lg–TDI, and AgNPs–Lg–TDI films embedded with AgNPs synthesized using lignin. For undoped TDI, UV radiation with wavelengths above 280 nm could not be filtered, while lignin–TDI (Lg–TDI) were not optically transparent in the UV-B region (280–315 nm) but were slightly transparent in the UV-A region (315–400 nm). These observations could be attributed to the abundance of aromatic rings, and the *p*–π conjugation of phenol, hydroxyl, and carbonyl functional groups in the lignin molecule, which enabled it to absorb large amounts of UV light [38]. Incorporation of AgNPs into the TDI matrix facilitated absorption of UV radiation with wavelengths between 280–360 nm. Presence of AgNPs within the Lg–TDI matrix enhanced UV-shielding efficiency of the films; hence, they can be used to block harmful UV radiation. Moreover, the films absorbed more radiation in the visible region as compared to TDI, which did not have lignin and silver embedded within the matrix [38]. Incorporation of lignin in the TDI matrix decreased the transmittance of UV radiation, attributed to the increased interaction between the OH of lignin and NCO groups in TDI during the synthesis of polyurethane.

### 4.2. FT-IR Spectra of Lg, Lg–TDI, AgNPs–Lg–TDI

The functional groups present in Lg, Lg–TDI, and AgNPs–Lg–TDI were determined using FTIR, and the results are depicted in Figure 6.

The hypothesis of this study was that isocyanate functional groups reacted with hydroxyl groups on the surface of lignin. Therefore, FTIR was performed to determine the functional groups involved. The IR spectra of Lg, Lg–TDI, and AgNPs–Lg–TDI are depicted in Figure 3. From Figure 3, lignin isolated from *Parthenium hysterophorus* depicted bands at 3442, 2937, 1713, and 1096 cm^−1^ vibrational frequencies, which were attributed to the presence of OH, CH_2_, C=O, and C–O–C vibrational bands. The IR spectra of lignin–TDI films displayed several bands at 2930 cm^−1^, 1680 cm^−1^, and 1450 cm^−1^, which correspond to the vibrational frequencies of the CH, N–H, and C–H groups, respectively [39,40]. Peaks below 1000 at 500 cm^−1^ were also reported. From the FTIR spectrum of the AgNPs–Lg–TDI, a narrow vibrational band at 3399 cm^−1^ and a band at 1536 cm^−1^ were attributed to -NH stretching vibrations, in-plane N–H bending vibrations, and C=O [11]. The absorption bands at 1637 and 1258 cm^−1^ were associated with NH_2_ deformation vibrations and the -C–N stretching vibrations, respectively [41]. The band due to the carbonyl stretching vibration (amide 1 band) occurs in the region between 1740–1680 cm^−1^. This range might slightly be lower in frequency depending on the solvent, but in the solid phase, primary urethanes have a broad C=O band that vibrates as low as 1690 cm^−1^ and it was present in our spectra [42]. The values at absorption bands of 1637, 1536, and 2930 cm^−1^ were a result of –C=O, –N=C=O, and –NH, which depicted the presence of isocyanate groups in surface functionalized lignin [43].

### 4.3. XRD Profile TDI, Lg–TDI, and AgNPs–Lg–TDI

The XRD patterns of AgNPs–Lg–TDI are presented in Figure 7.

The amorphous nature of lignin can be observed by the presence of a broad peak at 20° which corresponds to lignin’s amorphous character, and it is often observed for most, if not all, Lg–TDIs [44]. While this peak might be confused with that of residual polysaccharide, it is worth noting that isolation of lignin using the organosolv process eliminates this possibility as most polysaccharides, such as cellulose, are insoluble in alcohols. Moreover, peaks associated with cellulose have extensively been reported to be at 2θ values of 16, 22, and 32° linked to its amorphous and crystalline structures [31,45]. In the case of lignin, peaks observed at 2θ = 27, 32, 46, and 66° have been reported and are associated with extractives and hard segment (lignin) crystallization (Appendix A). These peaks often disappear when the resultant lignin is dialyzed to remove the extractives before incorporation in composite materials, but the broad peak at 2θ = 20° is associated with the amorphous domains of lignin. The powder diffractograms of Lg–TDI had two broad peaks at 2θ = 10 and 20, with the intensity of the peak at 10° used as a measure of the degree of crystallinity of the resultant polyurethane. A highly crystalline structure leads to formation of rigid composite, while soft composite exhibits a decrease in crystallinity evidenced by the disappearance of the peak centered at 10° [40]. These peaks are assigned to the scattering from TDI chains with regular interplanar spacing. Incorporation of AgNPs within the TDI matrix (Figure 4) was confirmed by the peaks at 2θ = 38, 44, 55, and 58°, which correspond to the silver crystal planes (111), (200), (220), and (311), respectively, while the peaks at 2θ = 28, 33, and 47° correspond to AgCl planes (111), (200), and (202), which were formed together with Ag^0^ during synthesis of AgNPs [26,46].

### 4.4. DSC Thermograms of AgNPs–Lg–TDI

The DSC thermogram of TDI, Lg–TDI, and AgNPs–Lg–TDI is depicted in Figure 8.

From the DSC thermograms of neat TDI and AgNPs–Lg–TDI (Figure 8), the films had four degradation stages at 63, 271, 370, 463 °C and 66, 275, 369, 455 °C, respectively. The glass transition temperature of toluene diisocyanate (TDI) films has been reported to be 40 °C, but thermal properties of these films are usually enhanced through copolymerization with other polymers during synthesis. Due to its high molecular weight and presence of aromatic groups within its structure, lignin, which is stable at high temperatures, tends to enhance the thermal stability of the resulting block copolymer which, when blended with TDI groups, forms polyurethane [40,41]. From the DSC curve obtained in this study, the T_g_ of the neat TDI and AgNPs–Lg–TDI was observed at 63 and 65 °C, where a sudden drop in the amount of heat absorbed was observed [47]. From Figure 5, neat Lg–TDI films had three degradation stages centered at 60, 269, 370, and 458 °C, which can be attributed to presence of adsorbed water and degradation of lignin and TDI in the films [2,5,41].

### 4.5. TGA and DTGA AgNPs–Lignin–TDI

The thermal decomposition of Lg–TDI is an important determinant for the technological prospects of their application. TGA measurements were performed to evaluate the thermal stability of TDI, Lg–TDI, and AgNPs–Lg–TDI, and the results are depicted in Figure 9 and Figure 10.

The degradation temperatures ranging from temperatures T_1_ to T_5_ with a maximum temperature T_max_ and the residual ash (wt%) are depicted in Table 1. The weight loss of the Lg–TDI starts at a temperature of 214 °C and continues to a final maximum degradation temperature (T_max_) of 457 °C. The isocyanate film showed a greater thermal stability relative to the other films since its abrupt thermal degradation started at 217 °C and reached T_max_ at 463 °C. In Lg–TDI and AgNps–Lg–TDI, the initial weight loss (<10%) that occurred between 50–150 °C was associated with moisture loss, decarboxylation reaction, and cleavage of ether bonds (β–O–4) present in lignin [38]. Lignin is moderately stable at higher temperatures as it has an aromatic backbone, but it undergoes a major weight loss between 200–600 °C [44]. The decomposition peaks found at 300 °C and 345 °C in the DTGA thermograms were attributed to the weight loss as a result of degradation of the phenylpropane side chains present in lignin and the elimination of water and carbon dioxide. The peak between 400–500 °C corresponds to the decomposition of its aromatic rings and the ester bonds, and the rupture of carbon–carbon linkages between lignin structural units [2,5,41]. This decomposition has also been reported, where it was observed that lignin was composed in two stages: alkyl-ether linkages, dehydration and decarboxylation reaction, and elimination of carbon dioxide and water. The observed higher ash content in AgNPs–Lg–TDI composite is as a result of silver nanoparticles embedded in the film as compared to undoped polyurethane films [38,44]. The high char residue in AgNPs–Lg–TDI was associated with the high thermal resistance of AgNPs encapsulated in the blends, though lignin has also been reported to have to impart thermal stability to its nanocomposite [2,5,41].

### 4.6. SEM Micrographs of AgNPs–Lg–TDIs

Surface morphology and size of the AgNPs–Lg–TDI simple filled polymer films are depicted in Figure 11.

From SEM micrographs (Figure 11), it was observed that silver nanoparticles randomly embedded in Lg–TDI composite had sizes ranging between 50 and 300 nm. As such, it can be concluded that the polymer matrix can act as a stabilizing agent for AgNPs which hindered agglomeration of AgNPs [48]. Due to presence of chloride ion during synthesis, both AgCl and AgNPs were present as both spherical nanoparticles, and cubic structure was observed in the SEM micrographs. Moreover, both AgCl and AgNPs crystal structures were also observed in the XRD profile of the thin films which can be attributed to the abundance of chloride ions in plant extracts.

### 4.7. Antimicrobial Assay Results

The results for the antimicrobial activity of TDI, AgNP–Lg–TDI, and the antimicrobial standards used for comparison are shown in Table 2.

Previous researchers have reported that the AgNPs have the ability to attach themselves to the bacterial cell membrane, and in the process penetrate into the cytoplasm, where the silver ions penetrate through ion channels, denature ribosomes, and suppress expression of enzymes and proteins necessary for energy production, thus disrupting the cell, eventually leading to cell death. The antimicrobial activity exhibited by the AgNPs–Lg–TDI, even though not very significant when compared to the blank, still offers possibilities for industrial and medical applications where films with antimicrobial potential are vital [49,50]. Incorporation of silver nanoparticles within a composite matrix enhances their antimicrobial activity due to the size and surface area of the nanoparticles, which enable them to penetrate the microbial cell wall [26]. Composite materials with lignin-based polyurethane as matrix and silver nanoparticles as filler shows that nanoparticles with antimicrobial activity can be attached onto biodegradable carrier materials such as polyurethane. The low activity is likely due to polymerization during the formation of lignin–TDI, which may have resulted in a bulky polymer, leading to reduced ability of silver to penetrate through the microbial cell wall to display any significant therapeutic activity.

## 5. Conclusions

Lignin extracted from *Parthenium hysterophorus* using the organosolv process was used to develop antimicrobial polyurethane films embedded with green synthesized silver nanoparticles. Being renewable in nature, it offers a promising alternative to petroleum-based polyurethane films. The excellent optical activity, thermal properties, and presence of AgNPs embedded within the polyurethane matrix implies that the films can be used as UV radiation blockers and as antimicrobial films. The AgNP–Lg–TDI did not have a significant antimicrobial activity against the test microorganisms but could still find applications in industrial and medical fields.

## Figures and Tables

**Figure 1 materials-16-04271-f001:**
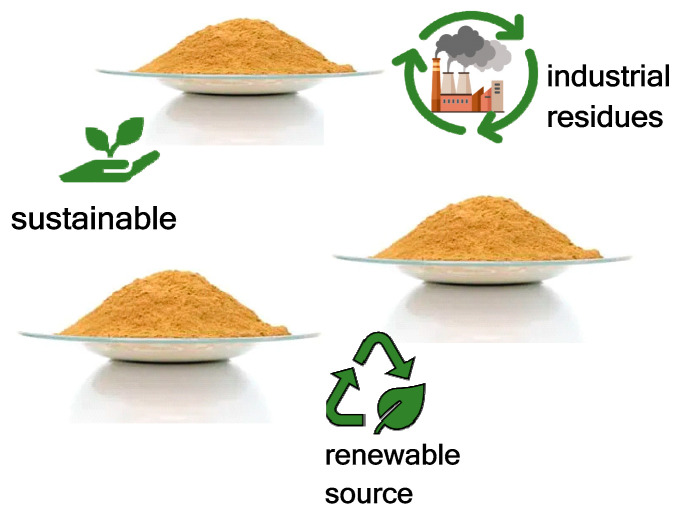
Lignin as a sustainable and renewable resource from different feedstocks.

**Figure 2 materials-16-04271-f002:**
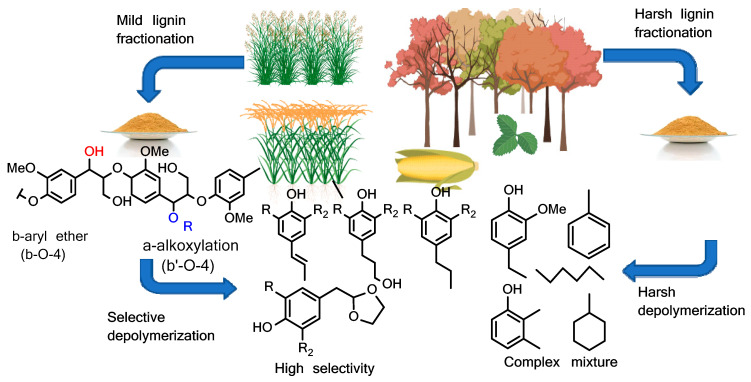
Pathways for isolation of lignin from agricultural biomass.

**Figure 3 materials-16-04271-f003:**
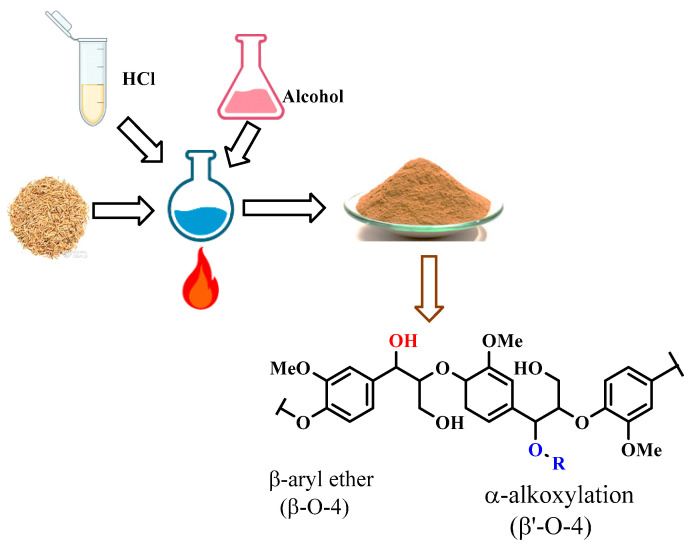
Extraction of organosolv lignin for use in Lg–TDI composite.

**Figure 4 materials-16-04271-f004:**
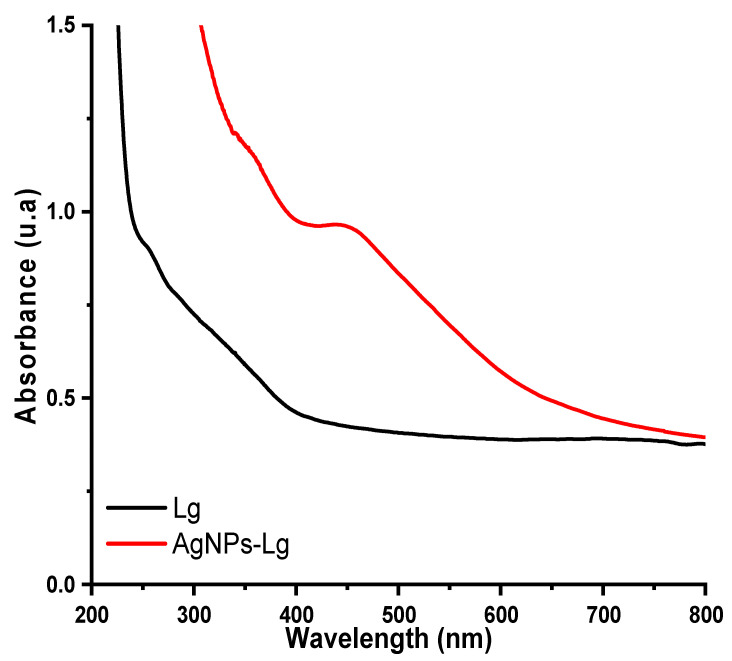
UV–Vis spectra of lignin and AgNPs–Lg particles.

**Figure 5 materials-16-04271-f005:**
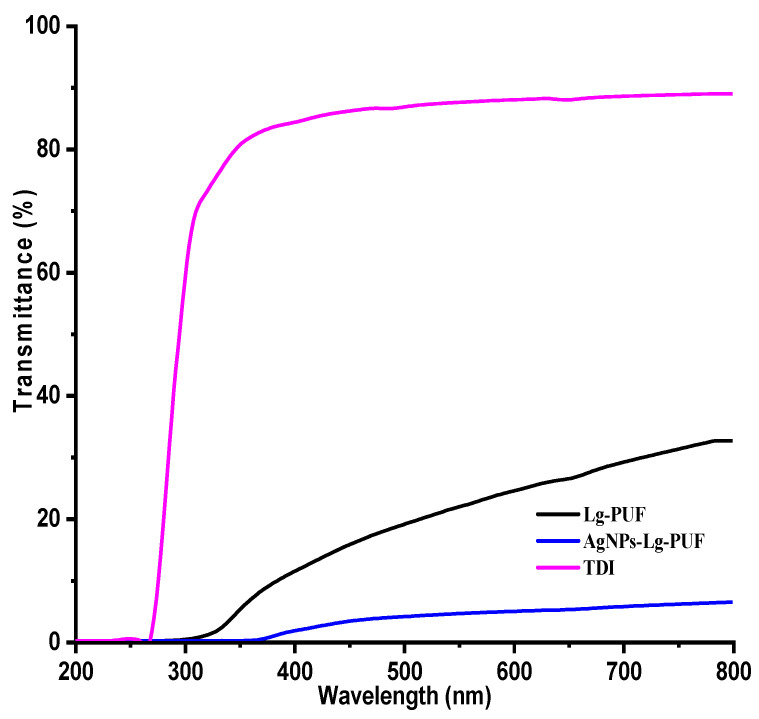
Evaluation of optical properties of TDI, Lg–TDI, and AgNPs–Lg–TDI.

**Figure 6 materials-16-04271-f006:**
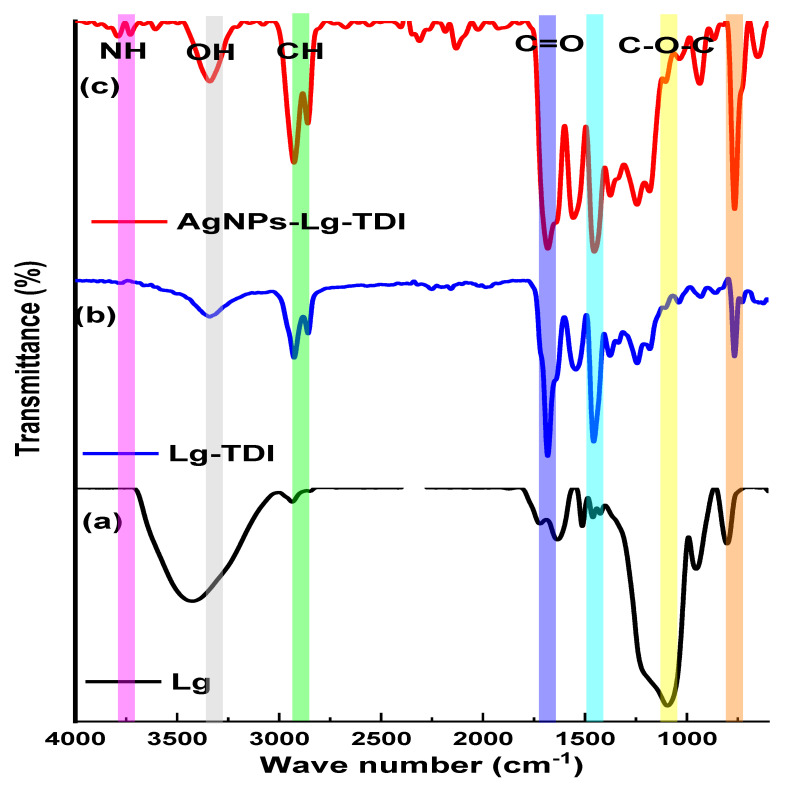
IR spectra of (**a**) Lg, (**b**) Lg–TDI, and (**c**) AgNPs–Lg–TDI.

**Figure 7 materials-16-04271-f007:**
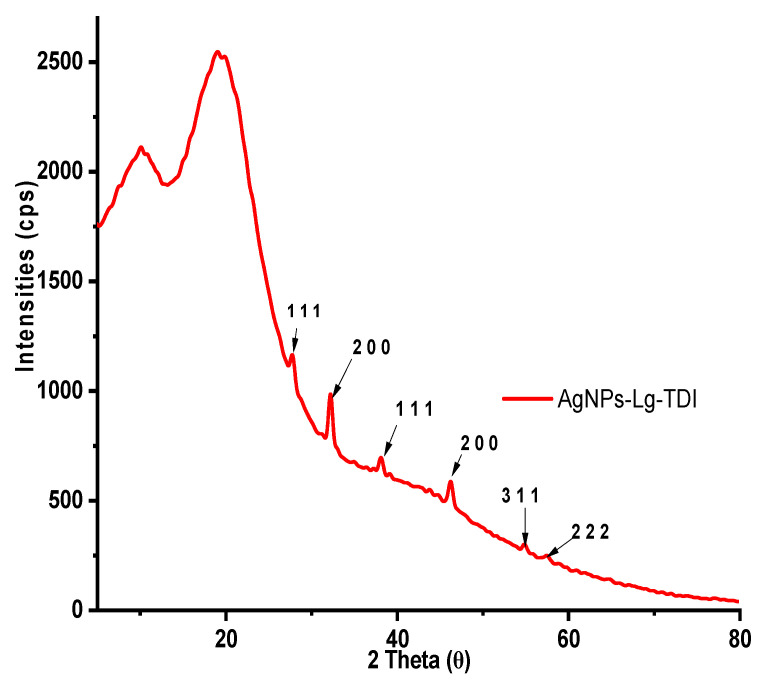
WXRD profile of AgNPs–Lg–TDI.

**Figure 8 materials-16-04271-f008:**
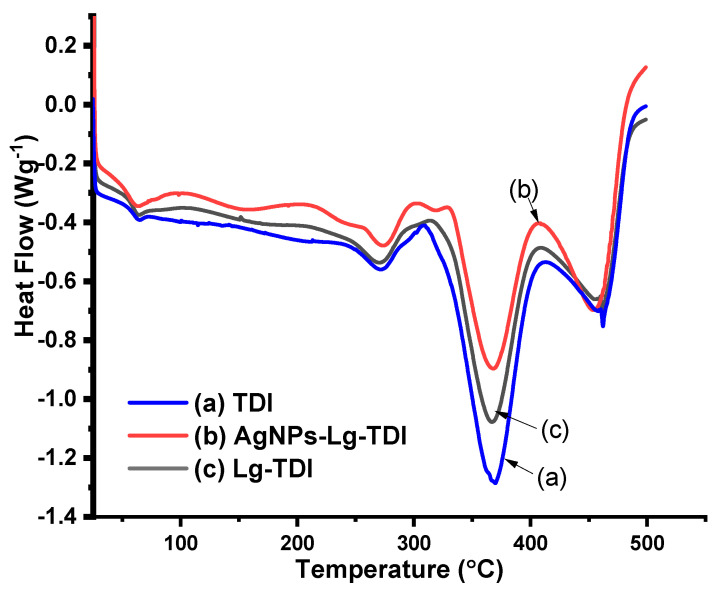
DSC thermogram of (**a**) toluene diisocyanate (TDI), (**b**) Lg–TDI, and (**c**) AgNPs–Lg–TDI film.

**Figure 9 materials-16-04271-f009:**
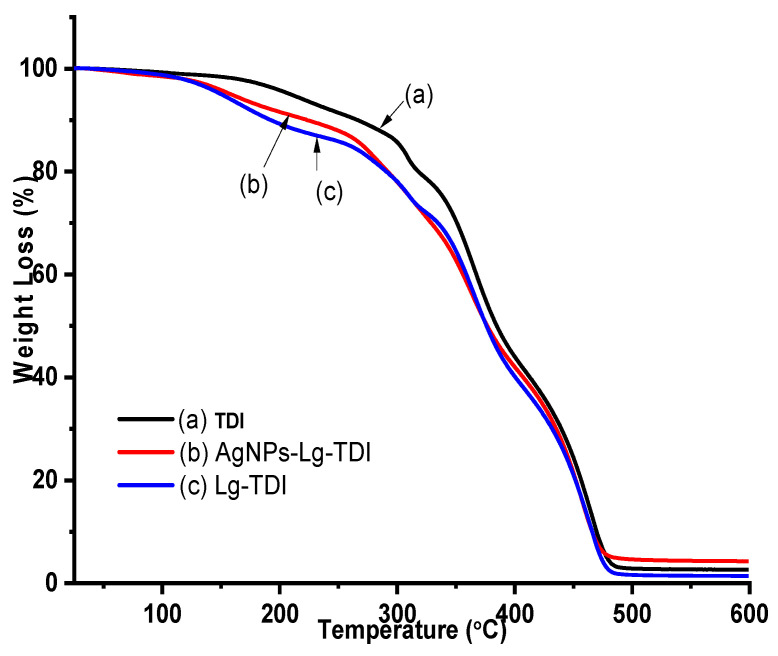
TGA thermogram of (**a**) toluene diisocyanate, (**b**) Lg–TDI, and (**c**) AgNPs–Lg–TDI film prepared using different solvents.

**Figure 10 materials-16-04271-f010:**
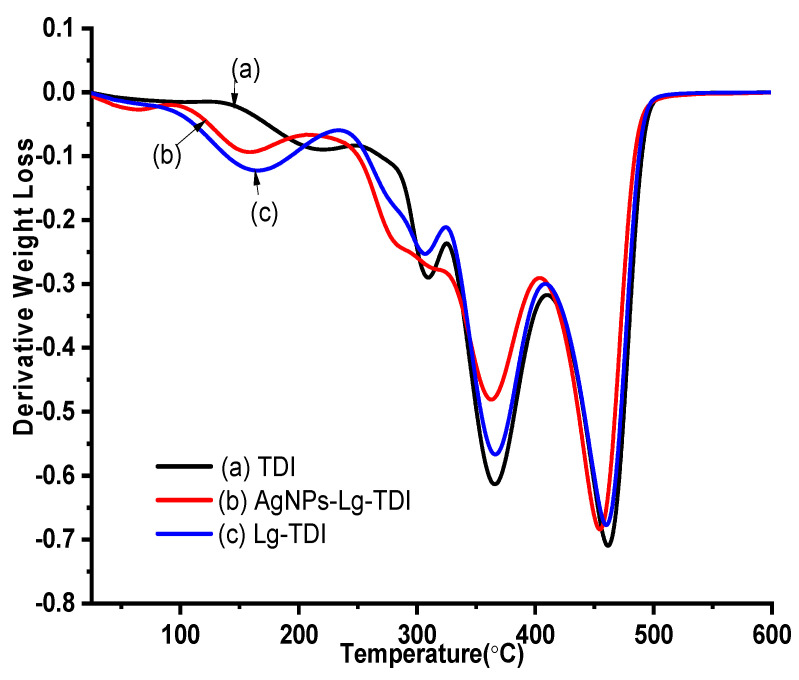
DTGA curves of (**a**) toluene diisocyanate, (**b**) Lg–TDI, and (**c**) AgNPs–Lg–TDI film.

**Figure 11 materials-16-04271-f011:**
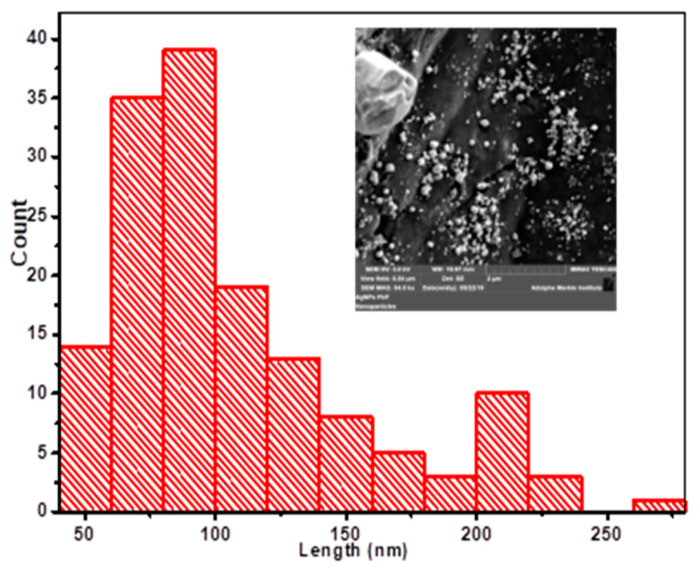
Size distribution of AgNPs embedded in lignin-based TDI composite film. Inset: SEM micrographs of AgNPs–Lg–TDI composite films.

**Table 1 materials-16-04271-t001:** Characteristic thermal degradation temperatures and residual ash (wt%) at 600 °C.

Sample	T_1_ °C	T_2_ °C	T_3_ °C	T_4_ °C	T_5_ °C	T_max_ °C	Ash (%)
Isocyanate	64		217	310	367	463	2.3
Lg–TDI	64	166	214	308	367	457	1.1
AgNPs–Lg–TDI	60	159	364	456			4.0

**Table 2 materials-16-04271-t002:** Zone of inhibition for TDI, AgNPs–Lg–TDI, and standard antibiotics.

	Minimum Inhibition Zone (Average ± Standard Deviation (mm))				
Sample	EC	PA	SA	BS	CA
Blank	7.0 ± 0.0	7.0 ± 0.0	7.0 ± 0.0	7.0 ± 0.0	7.0 ± 0.0
TDI	10.2 ± 0.3	8.1 ± 0.1	8.4 ± 1.0	7.2 ± 1.0	-
AgNPs–Lg–TDI	9.2 ± 1.0	8.4 ± 0.1	7.0 ± 1.0	8.1 ± 0.1	7.0 ± 0.1
AMC	14.3 ± 0.1	NI	9.2 ± 2.0	8.2 ± 0.4	NI
NIT	12.3 ± 2.0	NI	NI	8.5 ± 0.5	NI
NA	15.4 ± 0.3	NI	NI	9 ± 0.3	NI
GEN	13.2 ± 0.5	9.4 ± 0.6	8.2 ± 0.8	8.0 ± 0.2	NI
NX	28.0 ± 0.3	14.3 ± 0.2	9.1 ± 0.2	12.3 ± 0.4	NI
OF	27.5 ± 0.2	11.4 ± 0.6	NI	9.0 ± 1.4	12.1 ± 0.4
CTR	25.1 ± 2.0	9.2 ± 1.0	32.2 ± 1.5	NI	NI
SX	17.2 ± 1.4	14.1 ± 0.4	8.2 ± 0.5	12.8 ± 0.6	9.1 ± 0.4

KEY: AgNP: silver nanoparticle; Lg–TDI: lignin–toluene diisocyanate film; EC: *Escherichia coli*; PA: *Pseudomonas aeruginosa*; SA: *Staphylococcus aureus*; BS: *Bacillus subtilis;* CA: *Candida albicans*; AMC: Amoxyclav (20/10 µg); NIT: Nitrofurantoin (200 µg); NA: Nalidixic acid (30 µg); GEN: Gentamicin (10 µg); NX: Norfloxacin (10 µg); OF: Ofloxacin (10 µg); CTR: Ceftriaxone (30 µg); SX: Sulphamethoxazole (25 µg); NI: no inhibition.

## Data Availability

The authors declare that the data supporting the findings of this study are available within the paper and its Appendix A.

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
