# Peer review of "Encapsulation of AgNPs in a Lignin Isocyanate Film: Characterization and Antimicrobial Properties"

_materials, 2023, doi:10.3390/ma16124271_

Round 1

Reviewer 1 Report

The authors have proposed composite materials with lignin-based polyurethane as matrix and silver nanoparticles as filler. However, the authors have not been able to demonstrate a clear advantage to compositing. For example, the report mentions antimicrobial properties, but this is a function of the silver nanoparticles and not a benefit of compositing. Also, TGA and DSC were used to evaluate thermal degradability, but no clear improvement in heat resistance was demonstrated. Mechanical evaluations such as stress-strain curves, which are useful in the evaluation of composite materials, have not been performed. In addition, there are some mistakes, as follows.

Figures 1 and 2 are duplicated in the text.

Lignin extraction methods are described in 2.2 (line 81-88), and a similar statement is found in 2.3 (line 98-108). Moreover, the details of the lignin extraction methods described in 2.2 and 2.3 are different.

Silver nanoparticles are synthesized, but the spectra are only in text and the actual spectra are not posted.

What is a TDI film? Toluene diisocyanate (TDI) has a melting point of 21°C,   difficult to make a film from TDI itself.

There are many types of notation that may indicate Toluene diisocyanate film, which is confusing, so unify them. (TDI, TDI film, Isocyanate film, Toluene diisocyanate(TDI), ISO, Toluene diisocyanate and Isoyanate).

Mixed notation of Fig and Figure.

Based on the Lambert-Beer law, the optical path length (film thickness) is not stated even though it is an important value when discussing transparency.

Figure 2 in the Supporting Information is not explicitly stated in the text.

No photos of the film's appearance.

The orange line is drawn at 500 cm-1 in the FTIR in Figure 3 but is not attributed in the text for peaks below 1000 cm-1.

In FTIR, only facts and attribution and not enough discussion.

For example, in the silver nanoparticles composite film, there is a peak at 3442 cm-1 that is stronger than in the other films that are claimed to be derived from NH, but what does this mean?

Mixed notation of LPUF and LgPUF.

It is stated that the glass transition temperature of TDI film is reported to be 40°C (line 239), but there are no references cited. Also, the measured value of TDI film is stated as 63°C, which is a gap between the measured value and the literature value

The font size of TDI in Figure 6 is smaller than the others.

LgPUF begins to decompose at 241°C and the maximum decomposition temperature is described as 452°C, which does not correspond to the notation in Table 1.

Please describe the definitions of T1 to T5 and Tmax in Table 1.

Are the DSC and TGA measurements taken in the air or in noble gases?

No indication of what Lg stands for

Author Response

Lignin Reviewers Reports

The authors have proposed composite materials with lignin-based polyurethane as matrix and silver nanoparticles as filler. However, the authors have not been able to demonstrate a clear advantage to compositing. For example, the report mentions antimicrobial properties, but this is a function of the silver nanoparticles and not a benefit of compositing.

Also, TGA and DSC were used to evaluate thermal degradability, but no clear improvement in heat resistance was demonstrated.

Mechanical evaluations such as stress-strain curves, which are useful in the evaluation of composite materials, have not been performed.

In addition, there are some mistakes, as follows.

Figures 1 and 2 are duplicated in the text.

We have updated all figures and corrected the duplicate parts.

Lignin extraction methods are described in 2.2 (line 81-88), and a similar statement is found in 2.3 (line 98-108). Moreover, the details of the lignin extraction methods described in 2.2 and 2.3 are different.

 The statement for lignin extraction has been updated.

Silver nanoparticles are synthesized, but the spectra are only in text and the actual spectra are not posted.

What is a TDI film? Toluene diisocyanate (TDI) has a melting point of 21°C, difficult to make a film from TDI itself.

The synthesis was achieved through use of a small ratio of the TDI to lignin.

There are many types of notation that may indicate Toluene diisocyanate film, which is confusing, so unify them. (TDI, TDI film, Isocyanate film, Toluene diisocyanate(TDI), ISO, Toluene diisocyanate and Isoyanate).

TDI

TDI film

Isocyanate film

Toluene diisocyanate(TDI)

ISO

Toluene diisocyanate

Isoyanate

Mixed notation of Fig and Figure.

The notation has been corrected and Figure is used in the entire manuscript.

Based on the Lambert-Beer law, the optical path length (film thickness) is not stated even though it is an important value when discussing transparency.

Figure 2 in the Supporting Information is not explicitly stated in the text.

Supporting information on figure 2 has been added in line 8 to 11 of page 2 as highlighted in yellow.

 No photos of the film's appearance. The orange line is drawn at 500 cm-1 in the FTIR in Figure 3 but is not attributed in the text for peaks below 1000 cm-1.

This has been stated in the FTIR Results and discussion.

In FTIR, only facts and attribution and not enough discussion. For example, in the silver nanoparticles composite film, there is a peak at 3442 cm-1 that is stronger than in the other films that are claimed to be derived from NH, but what does this mean?

the FTIR data has been analyzed and further information added.

Mixed notation of LPUF and LgPUF.

The typo is corrected in pages 8 line 7, 14 line 9, and 17 line 4.

It is stated that the glass transition temperature of TDI film is reported to be 40°C (line 239), but there are no references cited. Also, the measured value of TDI film is stated as 63°C, which is a gap between the measured value and the literature value

The font size of TDI in Figure 6 is smaller than the others.

TDI in figure has been corrected though not coloured.

LgPUF begins to decompose at 241°C and the maximum decomposition temperature is described as 452°C, which does not correspond to the notation in Table 1.

The maximum temperature is 457°C not 452. This is corrected and heighted in yellow.

Please describe the definitions of T1 to T5 and Tmax in Table 1.

T1 to T5 and Tmax have been defined just below table 1 as the degradation temperatures ranging from temperatures T1 to T5 and a maximum temperature Tmax

Are the DSC and TGA measurements taken in the air or in noble gases?

No indication of what Lg stands for

Lg stands for lignin as indicated in the abstract line 7.

Reviewer 2 Report

In this study, a nanocomposite has been develop based on silver nanoparticles lignin polyurethane films. The films can be used as UV radiation blockers and antimicrobial films. But the AgNP-LgPUF did not have a significant antimicrobial activity against the test microorganisms. It is suggested that it be published after making the following modifications.

1.     Please change the thickness of lines 81 to 88 to a uniform form.

2.     The FTIR spectrum is not shown completely in Figure 3, and please improve it.

3.     Do the axes of WXRD profile in Figure 4 have units? Please include them.

4.     Please change the Angle units() in lines 210 to 230 to the correct form(°).

5.     The figure of surface morphology in Figure 8 is not clear, and please replace it

6.     The AgNP-LgPUF does not have a significant antibacterial activity. And why can still be used in industrial and medical fields? Please explain it.

Author Response

In this study, a nanocomposite has been develop based on silver nanoparticles lignin polyurethane films. The films can be used as UV radiation blockers and antimicrobial films. But the AgNP-LgPUF did not have a significant antimicrobial activity against the test microorganisms. It is suggested that it be published after making the following modifications.

  1. Please change the thickness of lines 81 to 88 to a uniform form.

The lines are changed as shown in the text.

  1. The FTIR spectrum is not shown completely in Figure 3, and please improve it.

The FTIT Spectra are presented in figure 6 and has been improved as requested.

  1. Do the axes of WXRD profile in Figure 4 have units? Please include them.

the units for the x axis is degree (°) while the y axis is counts per seconds which is indicated as cps

  1. Please change the Angle units(⁰) in lines 210 to 230 to the correct form (°).

All the units for angle have been changed to (°)

  1. The figure of surface morphology in Figure 8 is not clear, and please replace it

  1. The AgNP-LgPUF does not have a significant antibacterial activity. And why can still be used in industrial and medical fields? Please explain it.

Can be used as a drug delivery system.

Reviewer 3 Report

This manuscript reports the incorporation of silver nanoparticles in a lignin-based polyurethane to impart antimicrobial properties.

The silver-containing films are referred to as "composites" but no evidence for composite formation is presented. Are these not simple filled polymers or blends? This terminology needs to be corrected.

The manuscript will need revision for accuracy, clarity and readability. Corrections are pencilled-in directly on pages of the manuscript attached. These are illustrative of the kinds of changes needed throughout. In rewriting, careful attention should be paid to the use of articles, tenses and proper sentence structure. Author's names and et.al. should be omitted. Superfluous phrases such as "in the literature" should be avoided. "React" is not a transitive verb. Whether or not a reaction occurs depends on thermodynamics/kinetics not on the action of an operator. "Reacting" should be "treating." A study can't "investigate." Investigation occurs in the laboratory. Results of the investigation are reported in the paper.

This manuscript can be strongly improved with some careful editing.

Author Response

Reviewer Three comments

The manuscript will need revision for accuracy, clarity and readability. Corrections are pencilled-in directly on pages of the manuscript attached.

These are illustrative of the kinds of changes needed throughout. In rewriting, careful attention should be paid to the use of articles, tenses and proper sentence structure.

Author's names and et.al. should be omitted.

The author names and et al have been omitted in the manuscript

Superfluous phrases such as "in the literature" should be avoided. "React" is not a transitive verb. Whether or not a reaction occurs depends on thermodynamics/kinetics not on the action of an operator.

"Reacting" should be "treating."

Reacting has been replaced by treating

A study can't "investigate." Investigation occurs in the laboratory. Results of the investigation are reported in the paper.

Investigate has been removed in the entire manuscript

The requested changes that were attached as Pdf files have also been incorporated within the manuscript.

Reviewer 4 Report

This is a well written manuscript. The authors focus on the incorporation of AgNPs into lignin-based polyurethane film, which may exhibit certain antimicrobial activity. The authors have elucidated material preparation and characterization methods, and gave an explicit discussion on most of result data. The reviewer has only a few concerns on the material characterization methods and data analyses.

1. What method did the authors use to obtain the size distribution of AgNPs in Fig. 8? Was this method included in the materials and methods?

2. For the analysis of antimicrobial properties, it is suggested for the authors to differentiate gram positive and gram negative bacterial strains used in the study to discuss the antibacterial activity? it appears that AgNPs has a wide antimicrobial spectrum although the antimicrobial activity is not high. Almost all the antibiotics as control in the study do not have such a wide spectrum.

3. Why does TDI film have a similar antimicrobial activity to AgNPs-LgPUF film? Is there any benefit to perform polymerization and incorporation to form a complicated composite instead of directly using TDI film?

4. How possible would AgNPs be released from the AgNPs-LgPUF film in the practical applications?  

5. Some typos are present, such as Line 283, "size distribution".

Author Response

This is a well written manuscript. The authors focus on the incorporation of AgNPs into lignin-based polyurethane film, which may exhibit certain antimicrobial activity. The authors have elucidated material preparation and characterization methods, and gave an explicit discussion on most of result data. The reviewer has only a few concerns on the material characterization methods and data analyses.

  1. What method did the authors use to obtain the size distribution of AgNPs in Fig. 8? Was this method included in the materials and methods.

The method has been added and is highlighted in yellow under surface analysis in the methodology section.

  1. For the analysis of antimicrobial properties, it is suggested for the authors to differentiate gram positive and gram negative bacterial strains used in the study to discuss the antibacterial activity? it appears that AgNPs has a wide antimicrobial spectrum although the antimicrobial activity is not high. Almost all the antibiotics as control in the study do not have such a wide spectrum.

Silver nanoparticles have a wide antimicrobial spectrum as a result of the increased surface area and this is explained in page 15 line 14 to 18.

  1. Why does TDI film have a similar antimicrobial activity to AgNPs-LgPUF film? Is there any benefit to perform polymerization and incorporation to form a complicated composite instead of directly using TDI film?

Polymerization offers a delivery system for the antimicrobial activity.

  1. How possible would AgNPs be released from the AgNPs-LgPUF film in the practical applications?  

The process involves slow release of the nanoparticles under wet conditions.

  1. Some typos are present, such as Line 283, "size distribution". The term has been changed by deleting distribution.

Round 2

Reviewer 1 Report

The authors did not answer my question for about a third. In particular, rudimentary mistakes such as TDI films that should be unified in expression have not been sufficiently corrected and are not worthy of publication.

Author Response

The Term TDI have been unified throughout the manuscript

Reviewer one comments

The manuscript will need revision for accuracy, clarity and readability. Corrections are pencilled-in directly on pages of the manuscript attached.

These are illustrative of the kinds of changes needed throughout. In rewriting, careful attention should be paid to the use of articles, tenses and proper sentence structure.

Author's names and et.al. should be omitted.

The author names and et al have been omitted in the manuscript

Superfluous phrases such as "in the literature" should be avoided. "React" is not a transitive verb. Whether or not a reaction occurs depends on thermodynamics/kinetics not on the action of an operator.

"Reacting" should be "treating."

Reacting has been replaced by treating

A study can't "investigate." Investigation occurs in the laboratory. Results of the investigation are reported in the paper.

Investigate has been removed in the entire manuscript

The requested changes that were attached as Pdf files have also been incorporated within the manuscript.

Reviewer 3 Report

This manuscript is improved but still needs some attention. In the 2nd sentence under Introduction "they" is missing between "since" and "can." "Solvent according to the literature described" should be "solent as previously described." "Reported in various papers" should be "reported." "Synergy" should be "cooperation." 

These kinds of things need to be corrected throughout.

Author Response

This manuscript is improved but still needs some attention. In the 2nd sentence under Introduction "they" is missing between "since" and "can." "Solvent according to the literature described" should be "solent as previously described."

The statement has been rewritten

"Reported in various papers" should be "reported.

The statement has been revised as advised

" "Synergy" should be "cooperation." 

The term synergy has been replaced with cooperation

These kinds of things need to be corrected throughout.

The manuscript has been corrected to meet reviewers’ remarks

Reviewer 4 Report

My comments have been well addressed.

Author Response

Noted
